# Polyalthic Acid from *Copaifera lucens* Demonstrates Anticariogenic and Antiparasitic Properties for Safe Use

**DOI:** 10.3390/ph16101357

**Published:** 2023-09-26

**Authors:** Mariana B. Santiago, Vinicius Cristian O. dos Santos, Samuel C. Teixeira, Nagela B. S. Silva, Pollyanna F. de Oliveira, Saulo D. Ozelin, Ricardo A. Furtado, Denise C. Tavares, Sergio Ricardo Ambrósio, Rodrigo Cassio S. Veneziani, Eloisa Amália V. Ferro, Jairo K. Bastos, Carlos Henrique G. Martins

**Affiliations:** 1Laboratory of Antimicrobial Testing, Institute of Biomedical Sciences, Federal University of Uberlândia, Uberlândia 38405318, MG, Brazil; mariana.brentini@ufu.br (M.B.S.); drsantos.vco@hotmail.com (V.C.O.d.S.); nagela_bernadelli.mg@hotmail.com (N.B.S.S.); 2Laboratory of Immunophysiology of Reproduction, Institute of Biomedical Science, Federal University of Uberlândia, Uberlândia 38405318, MG, Brazil; samuctx@gmail.com (S.C.T.); eloisa.ferro@ufu.br (E.A.V.F.); 3Nucleus of Research in Sciences and Technology, University of Franca, Franca 14404600, SP, Brazil; pollyanna.oliveira@unifal-mg.edu.br (P.F.d.O.); sauloozelin@hotmail.com (S.D.O.); ricardo.furtado@unifran.edu.br (R.A.F.); denisecrispim2001@yahoo.com (D.C.T.); sergio.ambrosio@unifran.edu.br (S.R.A.); rodrigo.veneziani@unifran.edu.br (R.C.S.V.); 4Faculty of Pharmaceutical Sciences of Ribeirão Preto, University of São Paulo, Ribeirão Preto 14040900, SP, Brazil; jkbastos@fcfrp.usp.br

**Keywords:** *Copaifera lucens*, *ent*-polyalthic acid, antibacterial, antibiofilm, antiparasitic, toxicity

## Abstract

This study aimed at evaluating the potential of *Copaifera lucens*, specifically its oleoresin (CLO), extract (CECL), and the compound *ent*-polyalthic acid (PA), in combating caries and toxoplasmosis, while also assessing its toxicity. The study involved multiple assessments, including determining the minimum inhibitory concentration (MIC) and minimum bactericidal concentration (MBC) against cariogenic bacteria. CLO and PA exhibited MIC and MBC values ranging from 25 to 50 μg/mL, whereas CECL showed values equal to or exceeding 400 μg/mL. PA also displayed antibiofilm activity with minimum inhibitory concentration of biofilm (MICB_50_) values spanning from 62.5 to 1000 μg/mL. Moreover, PA effectively hindered the intracellular proliferation of *Toxoplasma gondii* at 64 μg/mL, even after 24 h without treatment. Toxicological evaluations included in vitro tests on V79 cells, where concentrations ranged from 78.1 to 1250 μg/mL of PA reduced colony formation. Additionally, using the *Caenorhabditis elegans* model, the lethal concentration (LC_50_) of PA was determined as 1000 μg/mL after 48 h of incubation. Notably, no significant differences in micronucleus induction and the NDI were observed in cultures treated with 10, 20, or 40 μg/mL of CLO. These findings underscore the safety profile of CLO and PA, highlighting their potential as alternative treatments for caries and toxoplasmosis.

## 1. Introduction

The oral cavity comprises various microenvironments such as tooth surfaces and mucosal epithelium. In each of these oral cavity sites, one can find about fifty species of microorganism, each of these locations in the oral cavity will be inhabited by different bacterial species, due to the cellular tropism of each microbe, there are around a thousand species that are capable of colonizing and integrating the microbiota of the oral cavity. Changes in the resident microbial community lead to dysbiosis, affecting the composition of the community [1,2].

In the case of dental caries, a diet with excess carbohydrates and sugars promotes the production of extracellular polymeric substances (EPS), the matrix that integrates the bacterial biofilm. As a result, saliva cannot neutralize the pH, and the bacteria in the formed and strengthened biofilm by a rich matrix of EPS produce acids that demineralize the tooth enamel, leading to dental caries [2]. Bacteria belonging to the lactobacillus and mutans streptococci groups, such as *Streptococcus mutans*, *S. mitis*, *S. salivarius*, and *Lactobacillus paracasei*, are directly responsible for the development of caries [3,4].

In 2019, according to the Global Burden of Disease, Injuries and Risk Factors Study, caries in permanent teeth were the most prevalent health condition in adults, with an estimated 2 billion cases. In the same year, caries in deciduous teeth were also the condition that most affected children (aged 0 to 14 years old), with an estimated 0.5 billion cases [5].

Chlorhexidine has been used as a gold-standard antimicrobial agent against cariogenic bacteria. Still, its long-time use can cause undesirable side effects such as taste change, the greenish-brown coloration of the teeth, mucosal peeling, and stone formation, in addition to the development of antimicrobial resistance in the oral microbiota [6]. That is the reason why developing new therapeutic alternatives with biological properties, capable of combating these diseases safely, becomes important and necessary.

Toxoplasmosis, caused by the protozoa *Toxoplasma gondii*, is an endemic disease that affects both humans and warm-blooded animals worldwide [7,8]. Its transmission occurs horizontally, involving the ingestion of food or water contaminated with infective oocysts, as well as the consumption of infected raw food among intermediate hosts, or through blood transfusion and organ donation from infected patients; it can also occur vertically which happens during pregnancy. The infection is usually asymptomatic or with mild flu-like symptoms in healthy humans. However, cases of clinical importance occur in immunosuppressed and pregnant individuals [7,8,9].

Even though approximately 30% of the world’s population is infected with *T. gondii* [10,11], toxoplasmosis is considered a neglected tropical disease [12,13], mainly because its higher incidence occurs in developing countries such as Brazil [7,8,11]. The gold-standard treatment against toxoplasmosis uses the drugs pyrimethamine (PYR) and sulfadiazine (SDZ). However, the therapy requires the prolonged use of the drugs which can increase toxicity and present significant failure rates in treatment [14]. Therefore, developing new therapeutic agents capable of safely combating the disease is also necessary.

To this end, natural products have been sourced to develop new drugs [15]. Trees belonging to the *Copaifera* genus are native to tropical regions of Latin America and West Africa. The genus *Copaifera* belongs to the Fabaceae family and includes 72 species. More than 20 *Copaifera* spp. exist in the Brazilian territory, where they are popularly called “copaibeiras”, “pau d’oleo”, or “copaíbas” [16,17].

The scientific literature contains numerous reports on the pharmacological activities of *Copaifera* species, such as their anti-inflammatory potential, antitumor, antiproliferative, anthelmintic, antituberculosis, gastroprotective, chemopreventive, immunomodulatory, and other antibacterial actions [16,17]. Although natural compounds have been traditionally employed, their use ought to occur safely; many studies have reported that several plant species applied in herbal medicine exhibit mutagenic, carcinogenic, or toxic effects [18].

The literature has widely described the chemical composition of different oleoresins [19,20]. In general, oleoresin is a natural product of secondary metabolism and consists mainly of a mixture of diterpene acids (resin fraction) and sesquiterpenes (volatile fraction) [16]. Although significant differences in the chemical profile of the oleoresin are observed between different species and various individuals of the same taxon, the diterpenes belonging to the types of skeleton caurano, clerodano, and labdano can be identified in all oleoresins [21], such as copalic, hardwickiic, kaurenoic, and polyalthic acids [22,23]. Specifically, for *C. lucens*, it was found that *ent*-polyalthic acid is the main constituent of this class of metabolites [24].

The diterpene *ent*-polyalthic acid, *ent*-15,16-epoxy-8(17),13(16),14-labdatrien-19-oic acid, is known for its biological properties, including gastroprotective [25], anti-inflammatory [26], antibacterial [27,28], antifungal [27], antitumor [29], and muscle relaxant actions [30]. Consequently, this molecule demonstrates potential to assist in the development of alternative therapies capable of addressing significant clinical conditions.

With the purpose of bringing scientific evidence to inform these health issues, the present study aimed to evaluate the potential anticariogenic properties of the crude extract of *C. lucens* (CECL), *C. lucens* oleoresin (CLO), and its major compound *ent*-polyalthic acid (PA). Additionally, the potential antiparasitic properties of PA were studied and the toxicity of the samples CLO and PA was assessed.

## 2. Results

### 2.1. Anticariogenic Activity

The anticariogenic evaluation of planktonic cells from CLO, CECL and PA was performed by determining the minimal inhibitory concentration (MIC) and minimal bactericidal concentration (MBC). The results are shown in Table 1.

The CLO and PA values ranged from 25 to 50 μg/mL, while the CECL values were higher or equal to the highest evaluated concentration of 400 μg/mL. CLO showed bactericidal activity against *Enterococcus faecalis* and *S. sobrinus*, PA showed bactericidal activity against *E. faecalis*, *S. mutans*, *S. salivarius*, and *S. sobrinus*, whereas CECL showed bactericidal activity against *S. mitis*, *S. salivarius*, and *S. sanguinis*. The positive control chlorhexidine presented MIC/MBC ranging from 0.92 to 7.37 μg/mL.

It was observed that the isolated compound PA presented the same bactericidal results as CLO; as it is also one of the major compounds present in CLO, it was considered that the anticariogenic activity presented by CLO is due to PA, and this compound was chosen for the continuation of the assessment of inhibition activity in biofilm formation.

To evaluate the anticariogenic activity of PA in bacterial communities, the biofilm formation inhibition assay was performed. Thus, the minimum inhibitory concentration of biofilm (MICB_50_) and cell viability (Log_10_ CFU/mL) were determined. The cariogenic strains were capable of forming biofilms at 10^7^ CFU/mL after 24 h of incubation in a suitable atmosphere.

The PA antibiofilm activity (Figure 1) presented MICB_50_ values that ranged from 62.5 to 1000 μg/mL, and it was able to completely eliminate cell viability for the strains *S. salivarius* (500 μg/mL), *S. sobrinus* (1000 μg/mL), *S. mitis* (250 μg/mL), *S. mutans* (125 μg/mL), and *L. paracasei* (500 μg/mL). The antibiofilm activity of the drug chlorhexidine, the standard drug used in the treatment of caries, against the strains used in the present study was also evaluated. The results are shown in Figure 2.

### 2.2. Antiparasitic Activity

For the antiparasitic evaluation assays, PA did not interfere with the cell viability of BeWo cells at concentrations of 64 μg/mL and lower, only losing viability at high doses of 128 μg/mL and 256 μg/mL (Figure 3a). In the assay for the evaluation of intracellular proliferation of *T. gondii*, the percentage was quantified through the activity of β-galactosidase in viable parasites. The inhibitory potential of compound PA was assessed at concentrations ranging from 4 to 256 μg/mL, and the combination of SDZ (200 μg/mL) and PYR (8 μg/mL) was also tested. The percentages of proliferation observed in the treatments described were compared with the negative control, which consisted of only infected BeWo cells. The data were graphically represented (Figure 3b), and after analysis, it was possible to identify that compound PA at concentrations of 32 (*p* < 0.01), 64 (*p* < 0.01), 128 (*p* < 0.0001), and 256 μg/mL (*p* < 0.0001), as well as SDZ + PYR (*p* < 0.0001), significantly inhibited the intracellular proliferation of *T. gondii* compared to the negative control. PA obtained a cytotoxicity concentration of 50% (CC_50_) of 171.76 ± 7.725, and inhibition concentration of 50% (IC_50_) of 93.24 ± 1.395.

Because of the promising anti-*T. gondii* activities exhibited by PA, a reversibility test was conducted to evaluate the maintenance of the antiparasitic effects. The assay was also performed on BeWo cells to quantify the intracellular proliferation of *T. gondii* through β-galactosidase activity. As expected, PA at 64 μg/mL (*p* < 0.01) and SDZ + PYR at 200 μg/mL and 8 μg/mL, respectively (*p* < 0.0001), reduced intracellular parasite proliferation at 24 h of treatment (Figure 3c). Interestingly, PA (*p* < 0.001) and SDZ + PYR (*p* < 0.0001) treatments maintained their ability to control parasite growth even after 24 h of treatment removal in comparison with the control group (infected cells incubated with RPMI 1640 medium only) (Figure 3c).

### 2.3. Toxicity Assessment

The toxicity evaluation of the survival fraction of V79 cells after treatment with different concentrations of CLO are shown in Figure 4. A significant reduction in colony formation was shown in concentrations ranging from 78.1 to 1250 μg/mL when compared to the group negative control. There was no significant difference in colony formation at the lowest concentrations tested (from 4.88 to 39.0 μg/mL). Therefore, the concentrations of 10, 20, and 40 μg/mL were chosen for the genotoxicity assessment.

The binucleated micronucleated V79 cell frequencies and nuclear division indexes (NDI) after treatment with CLO are shown in Figure 5. No significant difference in micronucleus induction and in the NDI were observed between cultures treated with 10, 20, or 40 μg/mL of CLO compared to the negative control group, revealing absence of genotoxicity and cytotoxicity, respectively.

Table 2 shows the results obtained from the in vivo evaluation in Swiss mice of the genotoxic potential of CLO and PA. No significant differences in the frequencies of micronucleated poly-chromatic erythrocytes (MNPCEs) were observed between animals treated with the different doses of CLO or PA and the negative control, demonstrating the absence of genotoxic activity. No significant reduction in the percentage of polychromatic erythrocyte (PCEs) in total red blood cells was observed in any of the treatment groups compared to the negative control, demonstrating the absence of cytotoxicity of the different treatments under the conditions tested.

The in vivo toxicity assessment using the *C. elegans* model was employed to determine the lowest concentration capable of killing 50% (LC_50_) of the larvae over time. Figure 6 shows the toxicity evaluation of different concentrations of PA over a 72-h period. The LC_50_ of PA was determined at 1000 μg/mL after 48 h of incubation.

## 3. Discussion

To date, no article has been found in the literature that evaluated the anticariogenic activity of *C. lucens*. Therefore, this study was the first to address this topic.

Holetz, et al. [31] consider that an extract exhibits good antibacterial activity at concentrations below 100 μg/mL, moderate activity between 100 and 500 μg/mL, and weak activity between 500 and 1000 μg/mL; above 1000 μg/mL, the extract antibacterial activity is considered inactive. Based on the authors’ criteria, it can be concluded that CECL showed moderate antibacterial activity against the strains *S. mitis*, *S. salivarius*, *S. sanguinis*, and *S. sobriunus* (MIC 400 μg/mL). On the other hand, CLO displayed good antibacterial activity against all seven cariogenic strains (MIC 25 μg/mL) evaluated in the present study.

Nevertheless, because of the lack of studies evaluating the anticariogenic potential of *C. lucens*, it is not possible to directly compare the data obtained in the present study with other authors. There are reports of the same biological property in other *Copaifera* species. Abrão et al. [32] evaluated the anticariogenic potential of the oleoresin of *C. duckei* and found MIC values between 25 and 50 μg/mL against the same strains. All of these aforementioned data indicate that *Copaifera* spp. is a source with active antibacterial potential, specifically against cariogenic bacteria. Furthermore, they validate the results obtained in the present study, as the MIC values found for CLO fall within the concentration range reported in the literature for other oleoresins of *Copaifera* spp.

Abrão et al. [32] evaluated the antibacterial activity of *ent*-polyalthic acid isolated from *C. duckei* against the cariogenic bacteria used in the present study, and the authors reported MIC and MBC values ranging from 25 to 50 μg/mL. The PA isolated from *C. lucens* used in the present study also exhibited MIC and MBC values within the same concentration range as reported by the authors. Despite the differences in the source of origin and extraction method, the compound used by the authors and in the present study refers to the same molecule, and therefore shares the same chemical composition. As previously mentioned, PA is the major compound in CLO, representing approximately 69.8% of the total oleoresin composition [24]. Regarding the oleoresin of *C. duckei*, the scientific literature reports values ranging from 6.9% to 40.86% [33,34,35] of *ent*-polyalthic acid in its composition. Therefore, CLO appears to be a source where a higher concentration of PA can be obtained.

When analyzing the MIC and MBC results of CLO and PA, it is observed that both samples exhibit identical results against the same strains with the exception of *S. mutans* and *S. salivarius* bacteria. For CLO, the MIC value was 25 μg/mL, while PA showed an MIC of 50 μg/mL against these particular strains. However, it is important to note that the bactericidal concentration was found to be the same for both samples. Based on these results, it may be suggested that the antibacterial activity exhibited by CLO is attributed to its compound PA. Therefore, since PA has been shown as a promising active compound, the experiments were pursued using it.

Bacterial biofilms exhibit high resistance to antibiotics, and the way it embeds within its EPS matrix is considered a virulence factor associated with the development of chronic infections which can affect even immunocompetent individuals [36]. Studies estimate that the treatment required for effective elimination of these bacterial communities needs to be administered at concentrations 10- to 1000-fold higher than those needed to eliminate the bacteria in their planktonic form [37,38,39]. This propensity was confirmed in the present study, where the concentrations of PA required to inhibit 50% of biofilm formation by cariogenic bacteria ranged from 15.6 to 1000 μg/mL, while the MIC to inhibit planktonic bacteria was 25 to 50 μg/mL. Even at the highest concentration evaluated (2000 μg/mL), it was not possible to completely eliminate the cell viability of the *E. faecalis* and *S. sanguinis* strains within the biofilm.

Abrão et al. [32] evaluated the activity of *ent*-polyalthic acid in inhibiting biofilm formation by *L. paracasei* (ATCC 11578), *S. mutans* (ATCC 25275), and *S. sobrinus* (ATCC 33478) strains, with a standardized bacterial inoculum concentration of 1 × 10^6^ CFU/mL. They obtained MICB_50_ value of 3.12 μg/mL against *L. paracasei* and 50 μg/mL against *S. mutans* and *S. sobrinus*, completely inhibiting cell viability at concentrations of 50 μg/mL (*L. paracasei*) and 200 μg/mL (*S. mutans* and *S. sobrinus*). In the present study, using the same strains, PA exhibited MICB_50_ of 15.6 μg/mL against *L. paracasei*, 125 μg/mL against *S. mutans* and 250 μg/mL against *S. sobrinus*, with complete inhibition of cell viability observed at concentrations of 500, 125, and 1000 μg/mL, respectively. It is important to note that in the present study, the inoculum concentration used was 1 × 10^7^ CFU/mL, which is higher than that used by Abrão et al. [32]. This difference in inoculum concentration may explain the data obtained in the present study.

Considering neglected diseases, *Copaifera* spp. can be a source with biological properties against parasites. Santos et al. [24] evaluated the activity against the promastigote form of *Leishmania amazonensis* of nine oleoresins from *Copaifera* spp. and all samples showed significant differences in parasite inhibition. One of the species evaluated by the authors was the oleoresin of *C. lucens* (containing 69.8% of PA in its composition), which exhibited IC_50_ of 20 ± 0.9 μg/mL against *L. amazonensis* [24]. Izumi, et al. [40] evaluated the oleoresin of *C. lucens* in the inhibition of *Trypanosoma cruzi* in three different forms of the parasite. They found IC_50_ value of 10.0 ± 2 μg/mL against the amastigote form, 51.0 ± 1.4 μg/mL against the epimastigote form and 215.0 ± 21.2 μg/mL against the trypomastigote form. Mizuno, et al. [41] evaluated the antileishmanial and antitrypanosomal activity of polyalthic acid. The authors found IC_50_ values of 8.68 ± 0.33 μg/mL against *L. donovani* amastigotes and 3.87 ± 0.28 μg/mL against *T. brucei* [41].

The initial findings presented in the current investigation indicate that a significant portion of the antimicrobial properties exhibited by CLO may potentially be ascribed to the bioactive constituent PA. Moreover, the current literature has demonstrated the promising antiparasitic effects of CLO [24] and PA against protozoan parasites [41]. Considering all the aforementioned points and taking into account that CLO consists of an approximate 69.8% composition of PA [24], we decided to exclusively employ PA for evaluating its anti-*T. gondii* efficacy within the BeWo cell model, which serves as the host cell model for this assessment.

In this context, Teixeira, et al. [42] evaluated the inhibitory activity of *ent*-polyalthic acid from *Copaifera* spp. against *T. gondii* tachyzoites using human villous explants as an experimental model. The PA used by the authors reduced the viability of villous explant at concentrations of 256 and 512 μg/mL. Furthermore, the compound was able to significantly inhibit the proliferation of tachyzoites at concentrations of 64 and 128 μg/mL. In addition to the inhibitory capacity at these concentrations (64 and 128 μg/mL), the compound also maintained its antiparasitic action even upon the removal of the treatment after 24 h of incubation [42]. In the present study, conducted in BeWo cells, the antiparasitic activity was similar, where the reduction in cell viability occurred at concentrations of 128 and 256 μg/mL, and the inhibitory action on *T. gondii* tachyzoites proliferation was significantly lower at non-cytotoxic concentrations of 32 and 64 μg/mL. At a concentration of 64 μg/mL, the antiparasitic activity of PA was maintained after 24 h of withdrawal of treatment. Taken together, using a different experimental model, our data are in agreement with the current literature that has demonstrated the prominent anti-*T. gondii* activity of PA, highlighting the potential of this bioactive compound.

As already mentioned, *C. lucens* oleoresin is rich in diterpenes, especially PA and copalic acid (CA), which are the major compounds [16]. Considering the popular use and the region of occurrence, CLO can be useful against tropical diseases and bacteria, so further studies could help to understand the effectiveness and possible human use. Furtado, et al. [43] carried out an extensive work, aiming to investigate the genotoxic potential of oleoresin from six different species of the *Copaifera* genus in mouse bone marrow. As a result, the absence of cytotoxicity and genotoxicity were found, even at the maximum limits recommended by OECD guidelines (2000 mg/kg) [44]. These results taken together with those obtained in the present study demonstrate the absence of significant genotoxic risk of oleoresin in these species of *Copaifera*.

The literature is scant regarding bioassays using PA in isolate form although it is known to be highly cytotoxic in human tumor cells [45] and antimutagenic against 3-amino-1,4-dimethyl-5H-pyrido(4,3-B)indole (Trp-P-1) genetic damage by Ames test [46]. In addition, PA had a considerable antibacterial and antifungal effect in vitro owing to cytotoxicity, including a greater effect than most of its semi-synthetic derivatives with extra carboxyl groups which were also tested [27].

CA is considered one of the biomarkers for *Copaifera* genus and represents up to 11% of the constitution of the CLO, but it is still considered as the second most abundant compound [24].

Another novel finding presented in our study is the toxicity evaluation using the *C. elegans* model. The use of the nematode model for toxicity assessment has been implemented satisfactorily as it provides an in vivo system with low maintenance requirements and expresses homologs of approximately 80% of human genes, despite some limitations such as evolutionary distance from humans and lack of organs [47]. The toxicity of PA was evaluated over a period of 72 h, and it showed good performance. Below the LC_50_ value (1000 μg/mL), even after 72 h of exposure, the compound exhibited a non-toxic profile to *C. elegans*.

All the toxicity evaluation data presented here, both for CLO and PA, indicate that they have the potential to become alternatives in the fight against cariogenic bacteria and *T. gondii*, as they did not show toxicity at the concentrations where they exhibited these biological properties.

## 4. Materials and Methods

### 4.1. Plant Material and Polyaltic Acid

*Copaifera lucens* Dwyer oleoresin (CLO) and crude extract of *Copaifera lucens* (CECL), obtained from the leaves of the tree, were collected in Rio de Janeiro, Brazil, between August 2012 and May 2014; this was authorized by the Brazilian government through SISBIO (35143-1) and CGEN (010225/2014-5). The plant material was identified by Haroldo Cavalcante de Lima at the Rio de Janeiro Botanical Garden Herbarium (JBRJ) and is deposited there under identification number 474304. Pure *ent*-polyalthic acid (PA) was obtained according to the methodology reported by our research group [35].

#### 4.1.1. Obtainment of CECL

The leaves of *C. lucens* were subjected to a dehydration process in a circulating air oven maintained at 40 °C and subsequently comminuted using a knife mill. One hundred grams of the resulting powder was subjected to a maceration process using a solvent consisting of ethanol and water in a 7:3 ratio (1 L). The solvent was systematically percolated, filtered, and then replaced, with this sequence being repeated every 48 h for a total of three cycles. The residual aqueous-enriched extract was preserved at a temperature of −80 °C and subsequently subjected to lyophilization in a Liobras Inc. apparatus until complete desiccation, yielding a total of 28 grams of the crude extract.

#### 4.1.2. Obtainment of CLO and Isolation of PA

The oleoresin was acquired through the process of perforating the trunk using an auger. The harvested oleoresin underwent filtration and was subsequently subjected to various chromatographic procedures. The methods employed for the isolation of PA were based on previously established procedures as described by our research group [35]. In summary, a quantity of 10 grams of oleoresin was subjected to vacuum liquid chromatography using organic solvents, leading to the generation of seven distinct fractions. PA was selectively eluted in fractions 2 and 3 which were subsequently combined and subjected to crystallization in a hexane-acetone mixture, resulting in the isolation of 1.2 g of pure compound. The chemical structure of PA was elucidated through the utilization of ^1^H and ^13^C Nuclear Magnetic Resonance in CDCl_3_, and its structural characteristics were compared to existing literature data [48].

### 4.2. Anticariogenic Activity

#### 4.2.1. Bacteria Used

The strains used in the study came from the American Type Culture Collection (ATCC, Manassas, VA, USA). The cariogenic strains used were *Streptococcus mutans* (ATCC 25175), *S. mitis* (ATCC 49456), *S. sanguinis* (ATCC 10556), *S. sobrinus* (ATCC 33478), *Lactobacillus paracasei* (ATCC 11578), *Enterococcus faecalis* (ATCC 4082), and *S. salivarius* (ATCC 25975). For all assays performed the bacteria were incubated in Brain Heart Infusion agar (BHI), added with defibrinated sheep blood (5%) in a microaerophilia incubator for 24 h at 37 °C with 10% CO_2_, except for *E. faecalis* and *S. salivarius* which were incubated aerobically at 37 °C for 24 h.

#### 4.2.2. Minimal Inhibitory Concentration (MIC) and Minimal Bactericidal Concentration (MBC)

MIC was determined by the microdilution technique in 96-well microplates following the recommendations of the Clinical and Laboratory Standards Institute [49], with adaptations using resazurin to reveal bacterial growth [50]. Briefly, solutions of the CLO, CECL, and PA in a concentration of 1.0 mg/mL were prepared in dimethylsulfoxide (DMSO, Sigma Chemical Co., St. Louis, MO, USA) followed by dilution in broth until concentrations ranging from 0.195 to 400 µg/mL were obtained, with a final DMSO content of 5% (*v/v*).

The inoculum concentration was adjusted to each microorganism. For cariogenic bacteria, the final concentration was 5 × 10^5^ CFU/mL. The positive control used was chlorhexidine (Sigma) at concentrations ranging from 0.115 to 59 µg/mL. The bacteria were incubated under the conditions described above. After the period of incubation, a 10-µL aliquot was removed from each well of the microplate and seeded on agar for the evaluation of MBC. The agar was incubated under appropriate conditions and, afterwards, the presence and/or absence of bacterial growth was verified to determine the MBC. Then, 30 µL of an aqueous solution of resazurin (0.02%) was added to each microplate to verify the microbial viability. Resazurin serves as an oxidation-reduction indicator, facilitating the real-time assessment of microbial viability. In this context, the blue hue indicates the absence of microbial viability, while the red color signifies the presence of microbial viability, therefore determining the value of MIC [50]. The sample evaluated was considered as bactericidal when it exhibited the same value in both the MIC and MBC assays. In cases where the MIC value was lower than the MBC value, the samples were considered bacteriostatic [51]. The experiments were carried out in triplicate.

#### 4.2.3. Antibiofilm Activity

The antibiofilm activity was evaluated by determination of the MICB_50_ which was the lowest concentration of the sample that inhibited the formation of 50% or more of the biofilm [52]. The method used was microplate dilution, a methodology similar to that used for determining MIC [49], with modifications. The experiments were carried out for the strains that showed better MIC results, performed in triplicate, with results demonstrated graphically.

The method used to inhibit the formation of biofilm was similar to the MIC assay conducted for planktonic cells. Serial dilutions of the samples were prepared in the wells of a 96-well microplate, the final concentration ranging from 0.98 to 2000 µg/mL. Chlorhexidine at concentrations between 0.115 to 59 µg/mL was assessed as negative control. The inoculum was added with 100 µL of each strain at a concentration of 10^7^ CFU/mL; the bacterial strains in the absence of chlorhexidine or PA were used as positive control. The bacteria were incubated under the conditions described above. Subsequently, following the incubation period, the contents within each well were carefully aspirated and each well underwent a series of three washes using 200 µL of sterile Milli-Q water to eliminate any planktonic cells. The biofilm adhered to the wells was fixed by exposing it to 150 µL of methanol for a period of 20 min. Antibiofilm activity was measured by MICB_50_ determined by optical density (OD) and by counting the number of colony-forming units (Log_10_ CFU/mL).

Based on the procedures described by Sandberg et al. [53], OD was quantified in the biofilm by adding 200 µL of crystal violet (0.2%) to the microplate wells. After 15 min at room temperature, excess dye was removed with tap water and dried in air at room temperature. Next, 200 µL of acetic acid at 33% (*v/v*) was added to each well to re-solubilize the dye bound to the cells. After 30 min, the OD of the microplates was measured at 595 nm using a microtiter plate reader (GloMax^®^, Promega, Madison, WI, USA). The percentage of inhibition was calculated using the equation:$$1-\left(\frac{At595nm}{Ac595nm}\right)\times 100$$
where At595nm and Ac595nm are the absorbance values of the wells treated with the samples and the control, respectively [52].

The antibiofilm activity measured by counting the numbers of CFU was performed to assess cell viability. Briefly, after the incubation period, the entire volume was carefully aspirated from the wells of the microplate and washed with water to completely remove the non-adherent cells. Then, 200 µL of broth was added to each well and the microplate went through the sonication process so that the adhered cells are released by the vibrations. Dilutions of 10^−1^ to 10^−7^ were performed for all wells and 50 µL of each dilution was plated in BHI agar added with defibrinated sheep blood (5%), which were incubated as already described. After incubation, the colonies were counted and the results were expressed in Log_10_ CFU/mL and shown graphically. Selection of the best inoculum concentration and incubation time for the antibiofilm activity assay was accomplished by standardizing biofilm formation.

### 4.3. Antiparasitic Activity

#### 4.3.1. Cell Culture and Parasite Maintenance

Human trophoblast cells were purchased commercially from the ATCC and were maintained following the protocols previously described by Drewlo, et al. [54]. In brief, cell culture maintenance was carried out employing RPMI 1640 medium (Cultilab, Campinas, SP, Brazil), supplemented with 100 U/mL penicillin (Sigma), 100 μg/mL streptomycin (Sigma), and 10% fetal bovine serum (FBS) (Cultilab). The cultures were incubated at a temperature of 37 °C in a humidified environment containing CO_2_ (5%).

Tachyzoites of *Toxoplasma gondii* (virulent RH strain, 2F1 clone), which consistently expressed the β-galactosidase gene, were cultivated following established protocols as described elsewhere [55]. Briefly, tachyzoites were maintained by serial passages in BeWo cells cultured in a RPMI 1640 medium containing 2% FBS, 100 U/mL penicillin, and 100 μg/mL streptomycin under controlled conditions of 37 °C and 5% CO_2_.

#### 4.3.2. Viability of the Host Cell

The viability of the host cell in the presence of PA was evaluated to determine the non-toxic concentration of the compound. The viability of BeWo cells treated with different concentrations of PA was assessed by MTT colorimetric test as described by Mosmann [56]. Briefly, in a 96-well plate, BeWo cells were placed at a concentration of 3.0 × 10^4^ cells/well for adhesion, after PA solutions were added at concentrations ranging from 4 to 256 μg/mL. Tests using the solvent DMSO at 1.2% (percentage present in the highest concentration used) was also performed. Also, cells incubated with only culture medium were used as a positive control of cell viability.

Microplates were incubated for 24 h at 37 °C under a humidified atmosphere and 5% CO_2_. After this period, the supernatant was removed and 10 µL of MTT (5 mg/mL) plus 90 µL of supplemented RPMI 1640 was added; the microplates were again incubated as described above for 4 h, followed by the addition of 10% sodium dodecyl sulfate (SDS, Sigma) and 50% N,N-dimethyl formamide (Sigma) with further incubation for 30 min. MTT reduction was measured at 570 nm absorbance using a multi-well scanning spectrophotometer (Titertek Multiskan Plus, Flow Laboratories, McLean, VA, USA). The values obtained were expressed in percentage of cell viability (cell viability %), where the absorbance of cells incubated only with culture medium were considered 100% viable. Assays were performed with eight replicates and demonstrated graphically.

#### 4.3.3. Evaluation of Intracellular Proliferation of *Toxoplasma gondii*: β-Galactosidase Activity

Concentrations of PA were used to evaluate its effect modulating the growth of a highly virulent strain of *T. gondii* (RH strain, 2F1 clone), using BeWo cells as a host, and the β-galactosidase colorimetric assay as previously described [57]. For this purpose, BeWo cells at a concentration of 3.0 × 10^4^ cells per well were placed in a 96-well microplate and infected with *T. gondii* tachyzoites at a multiplicity of infection (MOI) of 3:1 (ratio of parasites per cell). After three hours of infection, the medium was removed and the washing process was performed to remove excess parasites that did not infect the cells; afterwards, PA was added in concentrations ranging from 4 to 256 μg/mL. The association of sulfadiazine (SDZ—Sigma) plus pyrimethamine (PYR—Sigma) was used, as gold-standard drugs, at concentrations of 200 μg/mL and 8 μg/mL, respectively. The concentrations of SDZ + PYR used have been reported as non-toxic to BeWo cells, but efficiently control the parasitism [58]. Infected BeWo cells were incubated with RPMI 1640 medium in the absence of any drug and used as negative control. The plates were then incubated for 24 h at 37 °C and 5% CO_2_. To quantify the intracellular proliferation of *T. gondii*, the colorimetric β-galactosidase assay was used. The number of intracellular tachyzoites was calculated in comparison with the standard production of free tachyzoites (ranging from 15.625 × 10^3^ parasites to 1 × 10^6^). The percentage of proliferation (% *T. gondii* proliferation) was performed in comparison with the negative control (which shows 100% proliferation). Assays were performed with eight replicates and demonstrated graphically.

#### 4.3.4. Reversibility Assay

To assess the maintenance of the antiparasitic effects of PA, the reversibility assay was carried out as previously described [59,60], with the modifications described below. In brief, *T. gondii* tachyzoites were inoculated into BeWo cells at a MOI of 3:1. After 3 h of invasion, the cells were washed to eliminate the unattached parasites and used this basic experimental design to test the following situations: (1) following a 3-h invasion period, the intracellular parasites were permitted to proliferate under the influence of PA (at a concentration of 64 μg/mL) and SDZ + PYR (at concentrations of 200 μg/mL and 8 μg/mL, respectively), or in the absence of treatment, with only the culture medium (referred to as the untreated group) for a duration of 24 h. (2) the intracellular parasites were cultivated under identical conditions as described in (1), and after a 24-h treatment period, the cells were subjected to a thorough washing process, the culture medium was replaced, and the parasites were allowed to continue proliferating for an additional 24 h, this time in the absence of any treatment.

Subsequently, the reversibility rate was quantified as a percentage (reversibility of treatment %) at the 24-h mark after discontinuation of the treatment, with reference to both the untreated group (considered as 100% reversibility) and the corresponding treatment condition at the initial 24-h treatment period (utilized as the baseline for comparison). The assessment of *T. gondii* intracellular proliferation was conducted through the employment of the β-galactosidase assay. Assays were performed with eight replicates and demonstrated graphically.

### 4.4. Toxicity Assessment

#### 4.4.1. In Vitro System-Test

To carry out the cytotoxicity and genotoxicity experiments, Chinese hamster lung fibroblasts (V79 cells) were used. The cells were maintained in monolayer in plastic culture flasks (25 cm^2^) with Eagle’s minimal essential medium (DMEM) plus Nutrient Mixture F-10 (HAM-F10) 1:1 (Sigma) supplemented with 10% FBS, antibiotics (0.01 mg/mL streptomycin and 0.005 mg/mL penicillin; Sigma), and 2.38 mg/mL HEPES (Sigma), at 37 °C with 5% CO_2_ atmosphere. Under these conditions, the average cell cycle time was 12 h and the cell line was used after the 4th passage. All the experiments were performed in triplicate.

The cytotoxicity was evaluated by the in vitro cell survival assay based on the ability of a single cell to grow into a colony. The clonogenic efficiency assay was used according to the protocol described by Franken, et al. [61]. The cells (5 × 10^5^) were treated with CLO concentrations ranging from 4.88 to 1250 µg/mL for 3 h. In addition, a positive (methyl methanesulfonate, MMS, 110 μg/mL; Sigma), a negative (without treatment) and a solvent control (Tween 80 1%; Synth) were included.

The cytotoxicity results obtained by clonogenic efficiency assay led to the selection of three CLO concentrations (10, 20 and 40 µg/mL), which were used in the evaluation of the genotoxic potential by the micronucleus assay. The positive (MMS, 44 µg/mL), negative and solvent controls (Tween 80, 1%) were included.

Cytotoxicity was assessed using the IC_50_ value (50% cell growth inhibition) as a response parameter, which was calculated with the GraphPad Prism program by plotting cell survival against the respective concentrations of the test compound. All experiments were repeated independently at least three times.

#### 4.4.2. In Vivo System-Test

For the experiment, male and heterogenic Swiss mice (*Mus musculus*), weighing 30–40 g, obtained from the central house of animals of the University of São Paulo, Ribeirão Preto, Brazil, were used. Animals were maintained in ventilated cages under controlled conditions of temperature (23 ± 2 °C), humidity (50 ± 10%), light/dark cycle (12/12 h), food and water *ad libitum*. The study protocol was approved by the Ethics Committee for Animal Care of the University of Franca (process no. 2014/014).

In vivo studies were conducted to assess the genotoxicity of CLO and PA by micronucleus test in mouse bone marrow [44]. Animals were randomly divided in groups containing five animals each. For the treatments, CLO samples were diluted in 5% Tween 80, and PA samples were diluted in 5% DMSO (Sigma). Single doses of CLO (125, 250 and 500 mg/kg) and PA (1, 10 and 20 mg/kg) were administered by gavage in corresponding groups. Negative (water), Tween 80 (5%), DMSO (5%), and positive (MMS, 40 mg/kg, intraperitoneal) controls were included. The procedures and analyses were performed according to MacGregor, et al. [62].

#### 4.4.3. Toxicity Assessment in *Caenorhabditis elegans*

The in vivo model using *C. elegans* was used to assess the toxicity of PA; the procedure was performed as previously described in the literature [63]. The mutant strain *C. elegans* AU37 was used, the nematodes were cultivated in plates containing Nematode Growth Medium (NGM) and *Escherichia coli* OP50 kept in biochemical oxygen demand (BOD) at 16 °C for 72 h to obtain nematode eggs. After obtaining the eggs, the NGM plates were washed with M9 buffer to remove the larvae and eggs from the plates and transferred to tubes. A bleaching solution (hypochlorite + NaOH) was added to kill the adult larvae to obtain only the eggs. The eggs were added to a new NGM plate and incubated in BOD for 24 h at 16 °C. After this period, the eggs developed to the L1/L2 larval stage, the NGM plate was washed again, and the supernatant was then added to a new NGM plate containing *E. coli* OP50 and incubated in BOD for 24 h at 16 °C, for the larvae to synchronize to the L4 stage.

Toxicity assessment was performed in flat-bottomed 96-well microplates. Briefly, about 10 to 20 larvae of *C. elegans* in larval stage L4 were added to each well containing PA concentrations ranging from 31.25 to 1000 μg/mL. Toxicity control of the solvent used (1% DMSO) and positive control of larvae (larvae + culture medium) were carried out. The microplates were incubated in BOD at 25 °C for 72 h. Toxicity assessment was performed by counting live and/or dead larvae every 24 h under an inverted microscope. Larvae that showed movement were considered as alive, and those that remained static after being touched were considered as dead. The lethal concentration (LC) capable of killing 50% of the larvae was calculated. Assays were performed in triplicate and demonstrated graphically.

### 4.5. Statistical Analysis

The antiparasitic results were demonstrated as means ± standard deviations (S.Ds.). All data were checked first for normal distribution. Significance differences were assessed by comparison with controls by use of either one-way analysis of variance (ANOVA), Tukey’s or Dunnett’s multiple comparisons post-tests for the parametric data. Statistical differences were considered significant at *p* < 0.05.

The results of the micronucleus assays were analyzed statistically by analysis of variance for completely randomized experiments, with calculation of F statistics and respective *p* values. In cases where *p* < 0.05, treatment means were compared with Tukey’s test, the minimum significant difference was calculated for α = 0.05.

## 5. Conclusions

Based on the results obtained in the present study, it can be concluded that CECL exhibits moderate anticariogenic activity, while CLO demonstrates good anticariogenic activity, possibly due to the presence of the active compound PA which shows similar MIC/MBC results against the evaluated bacteria. PA also exhibits antibiofilm propertis, being able to inhibit 50% of biofilm formation at low concentrations and completely eliminate cells at higher concentrations. The compound (PA) also shows antiparasitic properties, inhibiting the *T. gondii* intracellular proliferation and maintaining its action even after treatment removal. Furthermore, the absence of genotoxicity and cytotoxicity of CLO and its major compound, PA, was revealed under experimental conditions.

## Figures and Tables

**Figure 1 pharmaceuticals-16-01357-f001:**
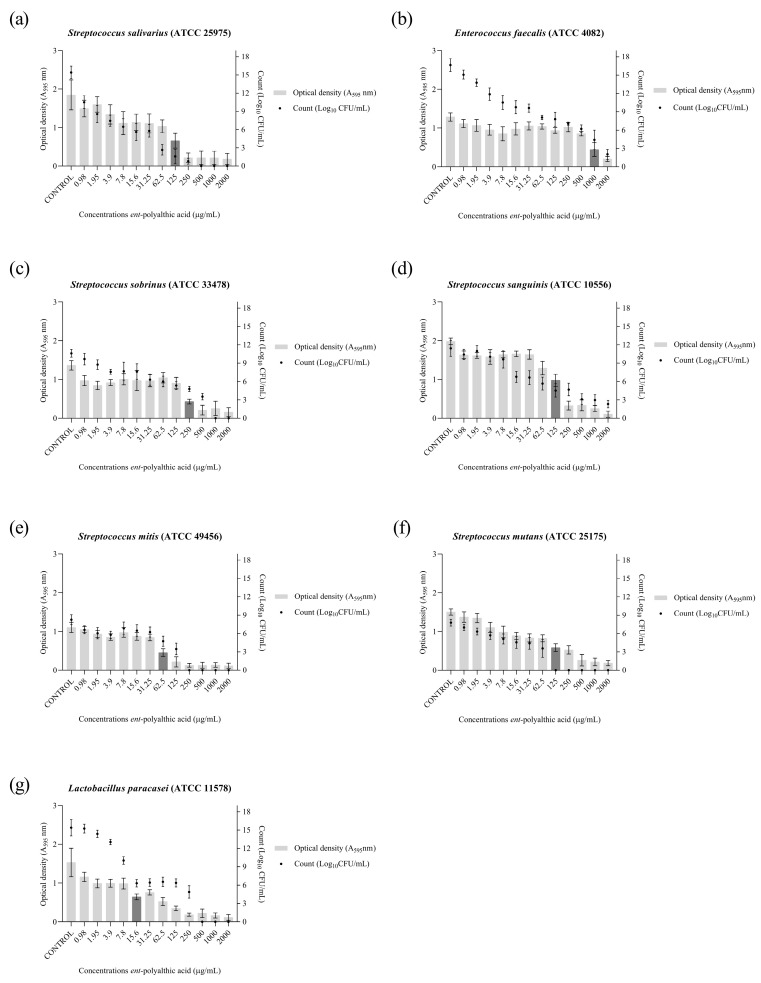
Graphical representation of antibiofilm activity as demonstrated by optical density (O.D.) and number of microorganisms (Log_10_ CFU/mL) of the *ent*-polyalthic acid against cariogenic bacteria. (**a**) *Streptococcus salivarius* (ATCC 25975); (**b**) *Enterococcus faecalis* (ATCC 4082); (**c**) *Streptococcus sobrinus* (ATCC 33478); (**d**) *Streptococcus sanguinis* (ATCC 10556); (**e**) *Streptococcus mitis* (ATCC 49456); (**f**) *Streptococcus mutans* (ATCC 25175); (**g**) *Lactobacillus paracasei* (ATCC 11578). The MICB_50_ value is represented in dark gray.

**Figure 2 pharmaceuticals-16-01357-f002:**
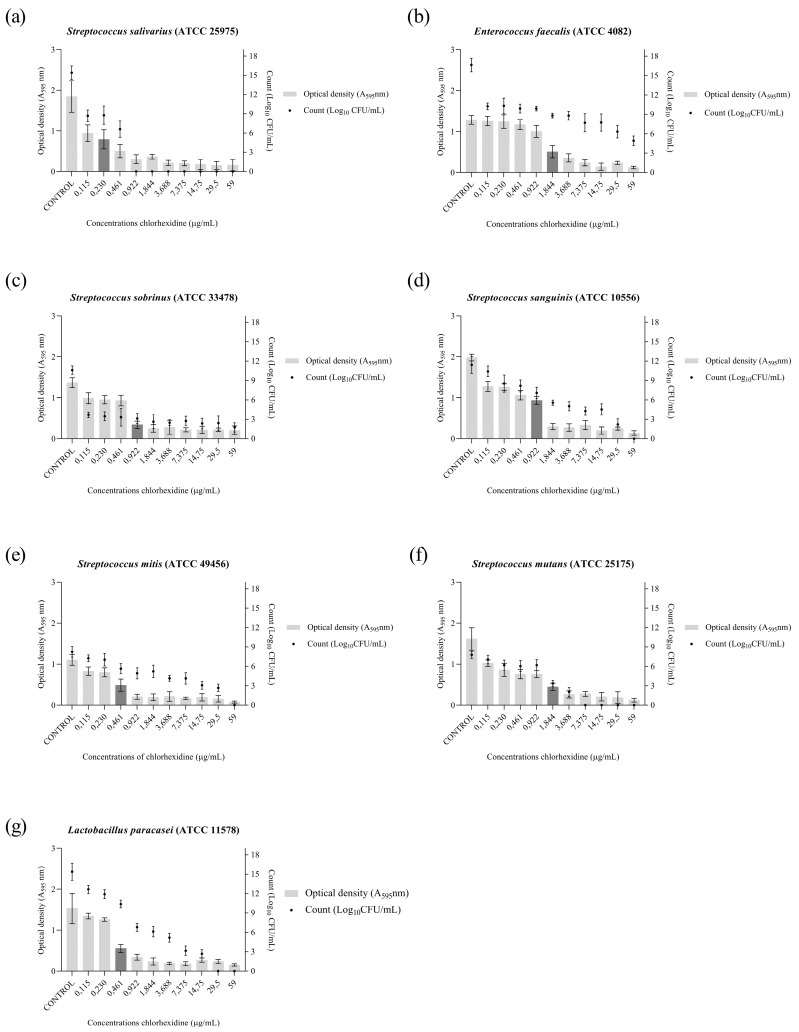
Graphical representation of the antibiofilm activity as demonstrated by optical density (O.D.) and number of microorganisms (Log_10_ CFU/mL) of the chlorhexidine against cariogenic bacteria. (**a**) *Streptococcus salivarius* (ATCC 25975). (**b**) *Enterococcus faecalis* (ATCC 4082). (**c**) *Streptococcus sobrinus* (ATCC 33478). (**d**) *Streptococcus sanguinis* (ATCC 10556). (**e**) *Streptococcus mitis* (ATCC 49456). (**f**) *Streptococcus mutans* (ATCC 25175). (**g**) *Lactobacillus paracasei* (ATCC 11578). The MICB_50_ value is represented in dark gray.

**Figure 3 pharmaceuticals-16-01357-f003:**
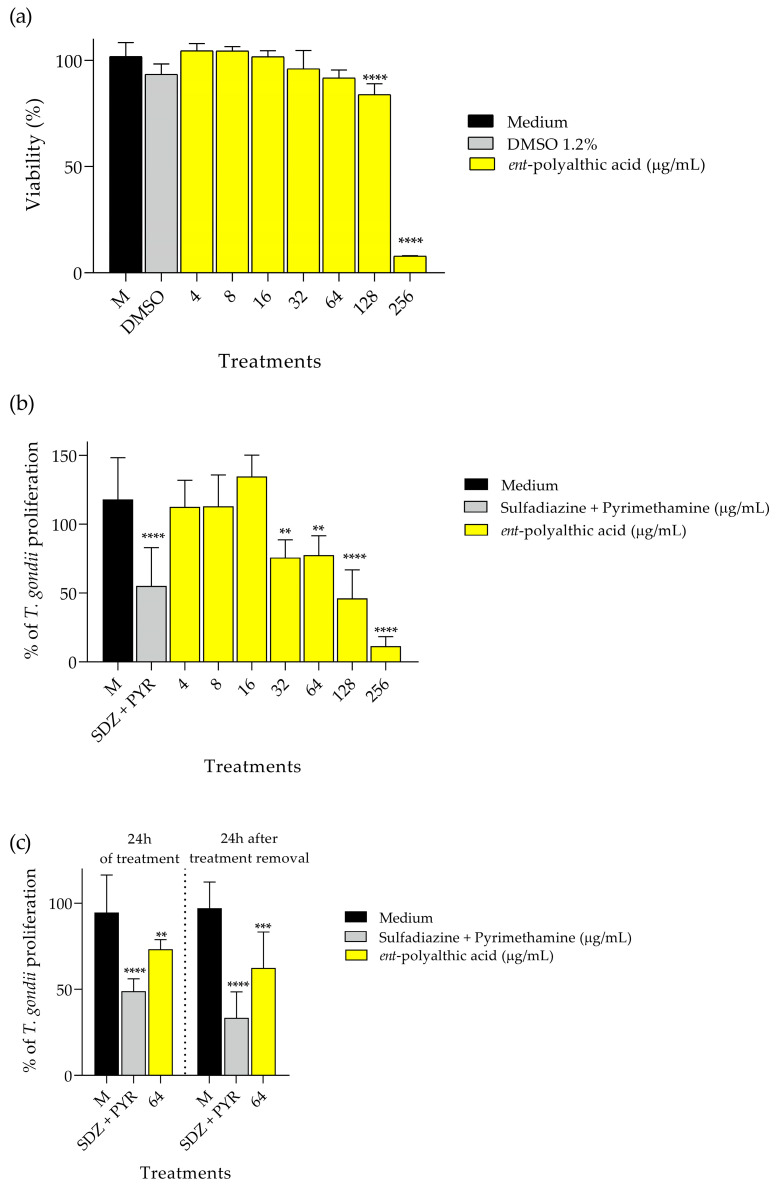
Graphical demonstration of the antiparasitic activity of the *ent*-polyalthic acid. (**a**) Viability of BeWo cells after 24 h of treatments. (**b**) *Toxoplasma gondii* intracellular proliferation after treatments. (**c**) Evaluation of the irreversibility antiparasitic action after treatment removal. ** Statistical significance with *p* < 0.01 when compared with the control (medium). *** Statistical significance with *p* < 0.001 when compared with the control (medium). **** Statistical significance with *p* < 0.0001 when compared with the control (medium).

**Figure 4 pharmaceuticals-16-01357-f004:**
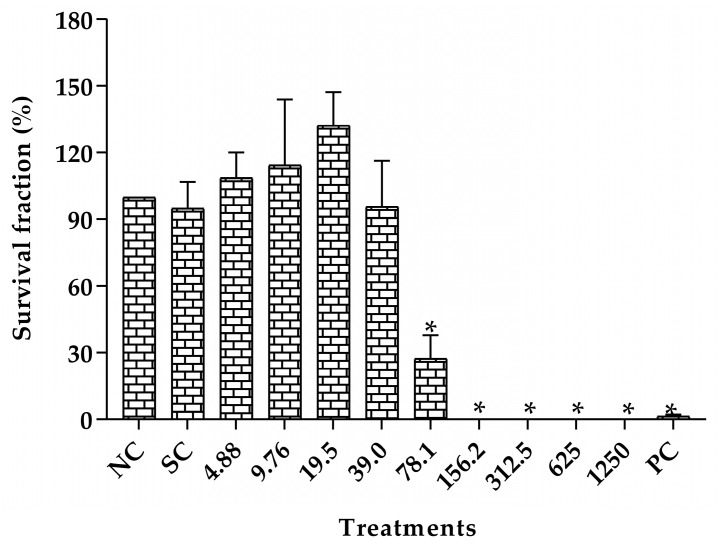
Survival fraction of V79 cells after the treatment with *Copaifera lucens* oleoresin. CLO (Copaifera lucens oleoresin), NC (negative control, without treatment), SC (solvent control, Tween 80, 1%), PC (positive control, methyl methanesulfonate, 110 µg/mL). * Significantly different from the negative control (*p* < 0.05).

**Figure 5 pharmaceuticals-16-01357-f005:**
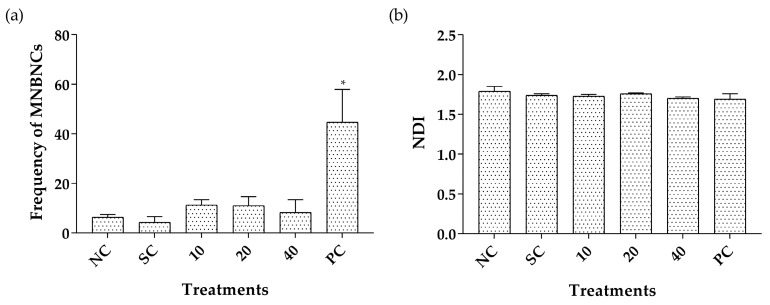
Binucleated micronucleated V79 cell frequency (**a**) and NDI (**b**) after treatment with *Copaifera lucens* oleoresin. CLO (*Copaifera lucens* oleoresin), NC (negative control, without treatment), SC (solvent control, Tween 80, 1%), PC (positive control, methyl methanesulfonate, 110 µg/mL). * Significantly different from the negative control (*p* < 0.05).

**Figure 6 pharmaceuticals-16-01357-f006:**
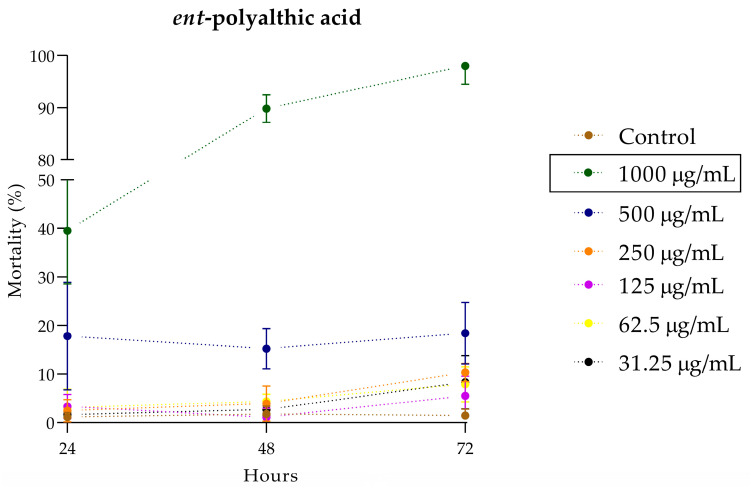
Evaluation of the toxicity of the *ent*-polyalthic acid in the *Caenorhabditis elegans* in vivo model.

**Table 1 pharmaceuticals-16-01357-t001:** Minimal Inhibitory Concentration (MIC) and Minimal Bactericidal Concentration (MBC) of *Copaifera lucens* oleoresin (CLO), crude extract of *Copaifera lucens* (CECL) an *ent*-polyalthic acid (PA) against cariogenic strains.

Cariogenic Strains	CLO (µg/mL)		CECL (µg/mL)		PA (µg/mL)		Chlorhexidine (µg/mL)	
	MIC	MBC	MIC	MBC	MIC	MBC	MIC	MBC
*Enterococcus faecalis* (ATCC 4082)	25	25	>400	>400	25	25	7.37	7.37
*Lactobacillus paracasei* (ATCC 11578)	25	50	>400	>400	25	50	3.68	3.68
*Streptococcus mitis* (ATCC 49456)	25	50	400	400	25	50	3.68	3.68
*Streptococcus mutans* (ATCC 25175)	25	50	>400	>400	50	50	0.92	0.92
*Streptococcus salivarius* (ATCC 25975)	25	50	400	400	50	50	0.92	0.92
*Streptococcus sanguinis* (ATCC 10556)	25	50	400	400	25	50	7.37	7.37
*Streptococcus sobrinus* (ATCC 33478)	25	25	400	>400	25	25	0.92	0.92

CLO—*Copaifera lucens* oleoresin. CECL—crude extract of *Copaifera lucens*. PA—*ent*-polyalthic acid. MIC—Minimal Inhibitory Concentration. MBC—Minimal Bactericidal Concentration.

**Table 2 pharmaceuticals-16-01357-t002:** Frequencies of MNPCEs and PCE/PCE + NCE ratio in the bone marrow of Swiss mice observed after the treatment with different doses of *Copaifera lucens* oleoresin, *ent*-polyalthic acid, and in respective controls.

Treatments (mg/kg)	MNPCEs	PCE/(PCE + NCE)
	Mean ± SD	Mean ± SD
Negative control	2.60 ± 1.34	0.59 ± 0.13
Tween 80	2.40 ± 1.14	0.72 ± 0.10
DMSO	1.60 ± 0.55	0.69 ± 0.02
CLO 125	5.00 ± 4.58	0.54 ± 0.04
CLO 250	9.60 ± 2.88	0.57 ± 0.07
CLO 500	6.80 ± 3.56	0.53 ± 0.07
PA 1	4.00 ± 0.55	0.56 ± 0.05
PA 10	3.60 ± 0.89	0.54 ± 0.02
PA 20	2.80 ± 1.79	0.56 ± 0.03
Positive control	34.60 ± 1.82 *	0.67 ± 0.14

MNPCEs—micronucleated polychromatic erythrocytes; PCE—polychromatic erythrocyte; NCE—normochromatic erythrocyte; Negative control—water; Tween 80, 5%; DMSO—dimethylsulfoxide, 5%; CLO—*Copaifera lucens* oleoresin; PA—*ent*-polyalthic acid; Positive control—methyl methanesulfonate, 40 mg/kg. *—Significantly different from the negative control group (*p* < 0.05).

## Data Availability

All data generated and analyzed during this study are available from the corresponding author upon reasonable request.

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
