# Peer review of "Polyalthic Acid from Copaifera lucens Demonstrates Anticariogenic and Antiparasitic Properties for Safe Use"

_pharmaceuticals, 2023, doi:10.3390/ph16101357_

Round 1
Reviewer 1 Report (Previous Reviewer 1)
The study by Santiago and colleagues reports the inhibitory activity of oleoresin and/or polyalthic acid (PA) on the growth of cariogenic bacteria and proliferation of Toxoplasma gondii tachyzoites in BeWo cells.
The activity of PA on cariogenic bacteria and T. gondii has already been reported in the literature, although it has been isolated from the species C. duckei. The same MIC values of oleoresin and polyalthic acid from both Copaifera species (C. duckei and C. lucens) were identified for cariogenic bacteria. Based on this, to improve the discussion, the authors should describe whether there is an advantage in working with C. lucens instead of C. duckei. For example, ease of obtaining oleoresin, geographic distribution of the plant, among other characteristics. As the concentrations of oleoresin and PA were the same, can the authors suggest the use of oleoresin based on economic criteria or other property?
In addition, other important issues must be clarified:
-Based on what criteria was it considered bactericidal only when MIC and MBC values ​​were the same? Please see the following review article: Levison ME. Pharmacodynamics of antimicrobial drugs. Infect Dis Clin North Am. 2004 Sep;18(3):451-65, vii. doi: 10.1016/j.idc.2004.04.012.
-The effect of phytochemicals on biofilm formation: how did the authors differentiate whether biofilm inhibition was due to the bactericidal effect of PA, or was there inhibition of adhesion and biofilm formation. According to the antibiofilm methodology, the authors describe that the test was similar to the one used to determine the MIC, except for the density of the bacterial inoculum.
-Statistical analyzes must be reviewed and significant differences must be properly reported. For example, in Figure 3, the percentage of viable cells after treatment with 128 ug/mL PA does not appear to differ significantly from the untreated control. In addition, there is no indication of just an asterisk, as described in the caption. (**, ***, ****)
-In in vivo genotoxicity assay, the PA concentrations analyzed were 1, 10 and 20 ug/mL. For S. salivarius and S. mutans, the PA MIC was equal to 50 ug/mL. Are concentrations greater than 40 ug/mL genotoxic?
-Was the treatment of C. elegans with oleoresin from C. lucens carried out? The results can contribute to the discussion.
Minor comments
-Resazurin can measured the viability of bacterial cells.
-Figure 1 and 2: please revise the names “Streptococcus sobrinus” and “Streptococcus mitis”. In the figure captions, the letter f) corresponds to Streptococcus mutans.
-line 59: please change “flora” to “microbiota”
-line 273: “The formation of bacterial biofilms exhibits high resistance to antibiotics” In general, microbial biofilms exhibit reduced sensitivity to clinically used antimicrobials when compared to planktonic cells. It is not biofilm formation that exhibits high resistance, as described.
-line 547: “The cytotoxicity was evaluation by the in vitro cell survival assay” please revise.
Minor editing of English language required.
Author Response
RESPONSE TO REVIEWERS
Reviewer #1
“The activity of PA on cariogenic bacteria and T. gondii has already been reported in the literature, although it has been isolated from the species C. duckei. The same MIC values of oleoresin and polyalthic acid from both Copaifera species (C. duckei and C. lucens) were identified for cariogenic bacteria. Based on this, to improve the discussion, the authors should describe whether there is an advantage in working with C. lucens instead of C. duckei. For example, ease of obtaining oleoresin, geographic distribution of the plant, among other characteristics. As the concentrations of oleoresin and PA were the same, can the authors suggest the use of oleoresin based on economic criteria or other property?”.
Response: We thank the reviewer for this pertinent observation. It has been verified in scientific literature that the oleoresin of C. duckei presents a lower percentage of ent-polyalthic acid (PA) of 6.9% to 40.86% (Carneiro et al., 2017; Carvalho et al., 2005; Cascon et al., 2000) in its chemical composition compared to the percentage found in the oleoresin of C. lucens (approximately 69.8%) (Santos et al., 2008). It has been added to the text that the oleoresin of C. lucens appears to be a source where PA can be found in more abundant concentrations than in C. duckei.
References:
Carneiro, L.; Bianchi, T.; da Silva, J.; Oliveira, L.; Borges, C.; Lemes, D.; Bastos, J.; Veneziani, R.; Ambrósio, S. Development and Validation of a Rapid and Reliable RP-HPLC-PDA Method for the Quantification of Six Diterpenes in Copaifera duckei, Copaifera reticulata and Copaifera multijuga Oleoresins. Journal of the Brazilian Chemical Society 2017, doi:10.21577/0103-5053.20170195.
Carvalho, J.C.; Cascon, V.; Possebon, L.S.; Morimoto, M.S.; Cardoso, L.G.; Kaplan, M.A.; Gilbert, B. Topical antiinflammatory and analgesic activities of Copaifera duckei dwyer. Phytother Res 2005, 19, 946-950, doi:10.1002/ptr.1762.
Cascon, V.; Gilbert, B. Characterization of the chemical composition of oleoresins of Copaifera guianensis Desf., Copaifera duckei Dwyer and Copaifera multijuga Hayne. Phytochemistry 2000, 55, 773-778, doi:10.1016/s0031-9422(00)00284-3.
Santos, A.O.; Ueda-Nakamura, T.; Dias Filho, B.P.; Veiga Junior, V.F.; Pinto, A.C.; Nakamura, C.V. Effect of Brazilian copaiba oils on Leishmania amazonensis. J Ethnopharmacol 2008, 120, 204-208, doi:10.1016/j.jep.2008.08.007.
“Based on what criteria was it considered bactericidal only when MIC and MBC values were the same? Please see the following review article: Levison ME. Pharmacodynamics of antimicrobial drugs. Infect Dis Clin North Am. 2004 Sep;18(3):451-65, vii. doi: 10.1016/j.idc.2004.04.012”.
Response: We appreciate the reviewer's suggestion regarding the methodology. In the review conducted by Levison (2004), bactericidal activity is considered when a drug is capable of reducing bacterial density from 105 to at least 102 CFU/mL, bactericidal when it is no more than four-fold higher than the MIC, and bacteriostatic when it is many-fold higher than the MIC. In the present study, we chose to follow a different methodology in which the content of the MIC is seeded onto a new agar without the presence of the antibacterial agent. Subsequently, after incubation, the absence or presence of bacterial growth is verified, determining the exact concentration at which no bacterial growth occurs. When the bactericidal concentration is equal to the inhibitory concentration, a bactericidal effect is determined, and when the bactericidal concentration is greater than the inhibitory concentration, a bacteriostatic effect is determined (Nayak et al., 2023). We have made changes to the manuscript, including the employed interpretation and the addition of a reference to the methodology used.
References:
Levison, M.E. Pharmacodynamics of antimicrobial drugs. Infect Dis Clin North Am 2004, 18, 451-465, vii, doi:10.1016/j.idc.2004.04.012.
Nayak, A.; Sowmya, B.R.; Gandla, H.; Kottrashetti, V.; Ingalagi, P.; Srinivas, V.S.C. Determination and comparison of antimicrobial activity of aqueous and ethanolic extracts of Amorphophallus paeoniifolius on periodontal pathogens: An in vitro study. J Indian Soc Periodontol 2023, 27, 40-44, doi:10.4103/jisp.jisp_182_22.
“The effect of phytochemicals on biofilm formation: how did the authors differentiate whether biofilm inhibition was due to the bactericidal effect of PA, or was there inhibition of adhesion and biofilm formation. According to the antibiofilm methodology, the authors describe that the test was similar to the one used to determine the MIC, except for the density of the bacterial inoculum.”.
Response: We appreciate the reviewer for their insightful question. In our study, we employed the methodology for evaluating biofilm formation inhibition standardized by Wei et al., (2006). When we mention that the test is similar to the MIC assay, we are primarily referring to the preparation of the 96-well microplate. In essence, in this process, we add culture broth, concentrations of the tested sample and the bacterial inoculum. It is noteworthy that the selection of the bacterial inoculum concentration is not arbitrary. Before conducting the assay, we standardize the bacterial concentration at which strains are capable of forming biofilms. In the case of this study, this concentration was standardized at 107 CFU/mL. This concentration differs from that used for evaluating planktonic cells in the MIC, which was 5x105 CFU/mL; it was observed that at this concentration, the bacterial strains were not capable of forming biofilms after 24 hours of incubation in our assays. In addition to standardizing the bacterial inoculum concentration for biofilm formation, we performed an important step after the biofilm growth. The entire well content was removed, and the well was washed three times with Milli-Q water to remove planktonic cells, ensuring that only the biofilm adhered to the microplate was evaluated in the subsequent stages of the methodology.
With that said, it is important to highlight that the evaluated compound demonstrated bactericidal activity against planktonic cells at a concentration of 5x105 CFU/mL. However, further tests would be required to confirm whether it would also exhibit the same property against a bacterial concentration of 107 CFU/mL.
References:
Wei, G.X.; Campagna, A.N.; Bobek, L.A. Effect of MUC7 peptides on the growth of bacteria and on Streptococcus mutans biofilm. J Antimicrob Chemother 2006, 57, 1100-1109, doi:10.1093/jac/dkl120.
“Statistical analyzes must be reviewed and significant differences must be properly reported. For example, in Figure 3, the percentage of viable cells after treatment with 128 ug/mL PA does not appear to differ significantly from the untreated control. In addition, there is no indication of just an asterisk, as described in the caption. (**, ***, ****)”.
Response: We appreciate this timely observation from the reviewer. A new analysis was performed, particularly in Figure 3, and it was observed that the graphical representation had an unnecessary stretching of the Y axis in the three graphs of the figure in question, which gave the impression that there were no differences between the treatments. We made adjustments to the axis range, and we believe that this makes it easier to interpret the graph and visualize the differences. Additionally, we also agree that the statistical data were not presented correctly; we have made changes to present the P values represented by each asterisk (**, ***, and ****).
“In in vivo genotoxicity assay, the PA concentrations analyzed were 1, 10 and 20 ug/mL. For S. salivarius and S. mutans, the PA MIC was equal to 50 ug/mL. Are concentrations greater than 40 ug/mL genotoxic?”.
Response: The genotoxic potential of PA was evaluated in mice at doses of 1, 10 and 20 mg/kg of body weight. MIC values for S. salivarius and S. mutans obtained in vitro were equal to 50 µg/mL. It is noteworthy that, in contrast to in vitro experiments, in vivo studies involve factors such as metabolism, pharmacokinetics and DNA repair processes and contribute to the responses. Therefore, considering that the genotoxicity studies were carried out in vivo and the antibacterial evaluation was conducted in vitro, it becomes difficult to establish a dose/concentration association between the two experimental models.
“Was the treatment of C. elegans with oleoresin from C. lucens carried out? The results can contribute to the discussion”.
Response: Considering that ent-polyalthic acid represents approximately 69.8% of the composition of Copaifera lucens oleoresin (Santos et al., 2008), and the compound exhibited the same antibacterial activity as the oleoresin (suggesting it as the bioactive component), we decided to perform the toxicity assay in C. elegans using only the compound. The same criteria were used to select ent-polyalthic acid as the sample evaluated in the antiparasitic activity.
References:
Santos, A.O.; Ueda-Nakamura, T.; Dias Filho, B.P.; Veiga Junior, V.F.; Pinto, A.C.; Nakamura, C.V. Effect of Brazilian copaiba oils on Leishmania amazonensis. J Ethnopharmacol 2008, 120, 204-208, doi:10.1016/j.jep.2008.08.007.
Minor comments
-Resazurin can measured the viability of bacterial cells.
Response: Corrected.
-Figure 1 and 2: please revise the names “Streptococcus sobrinus” and “Streptococcus mitis”. In the figure captions, the letter f) corresponds to Streptococcus mutans.
Response: Corrected.
-line 59: please change “flora” to “microbiota”
Response: Corrected.
-line 273: “The formation of bacterial biofilms exhibits high resistance to antibiotics” In general, microbial biofilms exhibit reduced sensitivity to clinically used antimicrobials when compared to planktonic cells. It is not biofilm formation that exhibits high resistance, as described.
Response: Corrected.
-line 547: “The cytotoxicity was evaluation by the in vitro cell survival assay” please revise.
Response: Corrected.

Reviewer 2 Report (Previous Reviewer 2)
Dear Authors,
I received your corrected manuscript according to my observations. You accepted most of my rewordings and comments, but, unfortunately, I did not receive a letter to reviewer, in which you should clarify my suspicion of self-citation, according to the image attached in the first review. The existence of that published abstract, with the same theme as the current manuscript, and the lack of your comments/elucidation is a major flaw for me. As such, my decision is Reject.
English is fine.
Author Response
RESPONSE TO REVIEWERS
Reviewer #2
“Dear Authors,
I received your corrected manuscript according to my observations. You accepted most of my rewordings and comments, but, unfortunately, I did not receive a letter to reviewer, in which you should clarify my suspicion of self-citation, according to the image attached in the first review. The existence of that published abstract, with the same theme as the current manuscript, and the lack of your comments/elucidation is a major flaw for me. As such, my decision is Reject”.
Response: First and foremost, we are pleased to finally be able to address the plagiarism accusation raised by the reviewer. However, we are disappointed to know that our letter to the reviewers, which was mandatory to be included during our resubmission process, did not reach the reviewer. We took the accusation very seriously, and we would like to clarify that the abstract in question is not from a published article (as indicated by the reviewer) but rather an abstract presented at an event (X Encontro PIBIC-PIBIC/EM-PIBITI-PIBI) held at the University of Franca in 2017 and published in its proceedings, as shown in the images below (site with all proceedings: https://publicacoes.unifran.br/index.php/investigacao/issue/view/171). The participation of the scientific community in such events is of utmost importance for the continued scientific dissemination, expansion of knowledge and building of new multidisciplinary networks. The abstract published in the proceedings is a result of the scientific dissemination of partial data, and it does not contain sufficient data (e.g., methodologies and results) that would compromise the data presented in the current manuscript. However, even with the reviewer's decision to reject our manuscript, we are pleased that this plagiarism issue has finally been clarified, as it does not align with the ethical standards of our research group.
The imagens are attached.

Reviewer 3 Report (New Reviewer)
Dear authors
The MS entitled “Polyalthic acid from Copaifera lucens demonstrates anticariogenic and antiparasitic properties for safe use” was evaluated thoroughly. The MS contents and hypothesis are significant. This study aimed to examine the anticariogenic and antiparasitic properties of oleoresin (CLO), extract (CECL) and the compound polyalthic acid (PA) of medicinal plant, Copaifera lucens. The toxicology was also evaluated. The MS is well composed and written systematically. My observations and some suggestion have been provided below.
· The abstract should be minimized from the introductory words or experimental and the actual results obtained should be incorporated in concise manner like MBC, MIC, Toxicity, viability etc. with actual data.
· Line 377: how was the extract lyophilized? Incorporate brief procedure.
· Line 380. The method lacks reference and also the literature reference is needed for coinciding NMR data.
· Line 447, equation should be written separate with an equation number.
· Line 455-456 Correct needed.
· The ent-polyalthic acid, its structure and biological importance should be discussed in the introduction section. Introduction needs revision.
English language needs some refinement.
Author Response
RESPONSE TO REVIEWERS
Reviewer #3
“The abstract should be minimized from the introductory words or experimental and the actual results obtained should be incorporated in concise manner like MBC, MIC, Toxicity, viability etc. with actual data”.
Response: We agree with the reviewer that the abstract should include the key findings of the study. Due to the 200-word limit, we couldn't detail all the results, but it has been rewritten to include the main findings.
“Line 377: How was the extract lyophilized? Incorporate brief procedure”.
Response: A change was made to the paragraph and the requested information was added.
“Line 380: The method lacks reference and also the literature reference is needed for coinciding NMR data.”.
Response: Corrected.
References:
Carreras, C.R.; Rossomando, P.C.; Giordano, O.S. Ent-labdanes in eupatorium buniifolium. Phytochemistry 1998, 48, 1031-1034, doi:10.1016/s0031-9422(98)00155-1.
“Line 447: Equation should be written separate with an equation number”.
Response: Corrected.
“Line 455-456 Correct needed.”.
Response: Corrected.
“The ent-polyalthic acid, its structure and biological importance should be discussed in the introduction section. Introduction needs revision.”.
Response: We agree with the reviewer and have made the suggested change in the manuscript.

Round 2
Reviewer 1 Report (Previous Reviewer 1)
Dear authors,
Thank you for all replies.
Congratulations.
Reviewer 3 Report (New Reviewer)
Dear authors. I am satisfied with the modifications.
The English is OK.
This manuscript is a resubmission of an earlier submission. The following is a list of the peer review reports and author responses from that submission.
Round 1
Reviewer 1 Report
The study by Santiago and colleagues reports the antibacterial activity of oleoresin, leaves crude extract, and polyaltic acid (PA) obtained from Copaifera lucens on caries-causing bacteria. Furthermore, they evaluated the effect of PA on Toxoplasma gondii proliferation, and in vitro and in vivo toxicity of plant compounds.
After careful reading, I highlight the following points that need to be clarified:
-In my opinion, the title is not appropriate. Throughout the text, the authors use the term anticariogenic to refer to the inhibitory effect of plant compounds on bacterial species that cause dental caries. Although the inhibitory effect on the bacteria that cause caries may have a direct effect on caries formation as a consequence, the term antibacterial/antimicrobial is more appropriate. I suggest change “anticariogenic” to “antibacterial activity against cariogenic bacteria” throughout the manuscript. In addition, the term “antitoxiplasmosis” is also not adequate. ...”antimicrobial activities against cariogenic bacteria and Toxoplasma gondii”
-Methods: The description of some methodologies is confusing.
a) How was the leaf crude extract obtained?
b) “Then, 30 µL of an aqueous solution of resazurin (0.02%) was added to each microplate to verify the microbial viability, therefore determining the value of MIC [42]”. Does this sentence describe how the MIC results were obtained?
c) The description of the antibiofilm activity needs to be improved. It is not possible to assess whether the compounds were added concomitantly to the bacterial inoculum. If so, what is the difference between the assay for MIC determination and this one for antibiofilm activity? Inoculum size only? As the compounds showed inhibitory activity on planktonic cells of the bacterial species tested, how is it possible to differentiate whether the antibiofilm effect occurred due to this inhibition? Or if it really inhibits the formation of biofilms?
d) According to the methodology described for evaluating the antibiofilm activity, the number of colony forming units was determined, not the number of microorganisms.
e) “Selection of the best inoculum concentration and incubation time for the antibiofilm activity assay was accomplished by standardizing biofilm formation”. Even if the standardization data of biofilm formation has not been shown, the size of the inoculum as well as the incubation time must be presented in the results.
f) There is no explanation why only PA was used in inhibitory assays for T. gondii.
g) What is the purpose of the reversibility test? In the description of the result, the way the test was performed is different from the description of the methodology. Please verify.
h) In toxicity assays, were CLO and PA not used in all in vitro and in vivo assays? Why?
-Results
The resolution of figures 1 and 2 needs to be improved.
The asterisks in the figures represent significant differences between which groups?
What do the x-axes of the figures 3, 4 and 5 represent?
Figure 3 c: reversibility test. If after removal of the medium containing the compounds, the T. gondii cells were incubated for an additional 24 h, would not an increase in cell number be expected? Mainly because the graph shows that there was no total inhibition of parasite proliferation. What would happen if the incubation time was increased?
-The discussion needs to be improved.
It is necessary to highlight the main findings of the study.
In the case of T. gondii, according to the discussion of the study, the antiproliferative activity of PA has been previously described. So, what is the contribution of the study presented to this topic?
The English language must be revised.
Minor comments
Lines 38-39: Please check the sentence content. “fifty species" “thousand species”?
Please, use “h” when referring to the time, even in the plural.
Please use “cell density” instead of “concentration”
Please use “minimal bactericidal concentration” instead of “minimal bacterial concentration”
Chlorhexidine is not an antibiotic. Please correct. (line 111).
The English language must be revised.
Reviewer 2 Report
Title
Line 3: The title: “ANTITOXOPLASMOSIS” instead of “ANTITOXIPLASMOSIS”!
Introduction section
Lines 36-37: I reworded it a bit: "The oral cavity comprises various microenvironments, such as tooth surfaces and mucosal epithelium. In each of these oral cavity sites, it is possible to find about ...".
Lines 44-46: Reworded: "As a result, saliva cannot neutralize the pH, and the bacteria in the formed and strengthened biofilm by a rich matrix of EPS, produce acids that demineralize the tooth enamel, leading to dental caries."
Lines 54-57: I divided your phrase into two shorter, more understandable sentences: "Chlorhexidine has been used as a gold-standard antimicrobial agent against cariogenic bacteria. Still, its use for a long time can cause undesirable side effects such as taste change, the greenish-brown coloration of the teeth, mucosal peeling, and stone formation, in addition to the development of antimicrobial resistance of the oral flora [6]."
Lines 60-61: Reworded: "Toxoplasmosis, caused by the protozoa Toxoplasma gondii, is an endemic disease that affects both humans and warm-blooded animals worldwide."
Lines 61-64: Contamination ways in toxoplasmosis are much more varied; as such, I reworded your sentence a bit: "Its transmission occurs horizontally, through ingestion of food or water contaminated with infective oocysts, consumption of infected raw food between intermediate hosts, or through blood transfusion and organ donation from infected patients, and vertically, which happens during pregnancy."
Lines 64-66: I rewrote your sentence: "The infection is usually asymptomatic or with mild flu-like symptoms in healthy humans. However, cases of clinical importance occur in immunosuppressed and pregnant individuals."
Lines 68-75: I rewrote this paragraph; now, it contains shorter sentences, avoiding using the same word twice in a sentence. I have also modified some expressions to eliminate unduly lengthening the manuscript.
"Even though approximately 30% of the world's population is infected with T. gondii [10,11], toxoplasmosis is considered a neglected tropical disease [12,13], mainly because its higher incidence occurs in developing countries, such as Brazil [7,8,11]. The gold standard treatment against toxoplasmosis uses the drugs pyrimethamine (PYR) and sulfadiazine (SDZ). However, the therapy requires prolonged use of the drugs, which can increase toxicity and present significant failure rates in treatment [14]. Therefore, developing new therapeutic agents capable of safely combating the disease is also necessary."
Line 78: It would be better to use the family name Fabaceae since "Leguminosae is an older name still considered valid and refers to the fruit of these plants, which are called legumes." [Wikipedia]
Line 83/84: "immunomodulatory" instead of "immunomodulator"
Line 84: use "Because..." instead of "Due to the fact that...". It seems as an unduly lengthening of the manuscript.
Lines 88-89: I rewrote your sentence for clarity: "The literature has widely described the chemical composition of different oleoresins [19,20]."
Line 89: "secondary metabolism" of who, or what? Please, specify!
Line 92: This: "between species and individuals of the same species" doesn't sound too scientific! If you refer to several species of the same genus, use "between different species and various individuals of the same taxon" to avoid repetitions.
Lines 97-101: Reworded: "To bring scientific news to these health issues, the present study aimed to evaluate the potential anti-cariogenic properties of crude extract of C. lucens (CECL), C. lucens oleoresin (CLO) and its major compound ent-polyalthic acid (PA). Additionally, we studied the potential antiparasitic properties of PA and assessed the toxicity of the samples CLO and PA."
Results section
Line 111: chlorhexidine is not an antibiotic considering its biochemical belonging; it is a disinfectant and antiseptic. Please, cut "antibiotic"
If you refer to the types of relationships according to the direct effects of species interaction, yes, in that case, chlorhexidine can be classified as an antibiotic because it intervenes in a relationship of antibiosis.
Line 113-116: this phrase is too long and less understandable. Please, reword it and divide it into two more easily understood sentences.
Lines 124-126: this paragraph includes two sentences, but I'm sure that you actually want to be only one sentence. Please check and reformulate! There is a wording mistake!
Line 150: "Antitoxiplasmosis activity"??? Or, better, "Antitoxoplasmic activity"?
Discussion section
Line 235: you used the "anti-cariogenic" wording, but I found "anticariogenic" in almost the entire manuscript. I'm sure the correct form is "anti-cariogenic"! So, please, change the wording throughout the manuscript!
Lines 235-236: your statement: "To date, no article has been found in the literature that evaluated the anti-cariogenic activity of C. lucens. Therefore, this study was the first to address this topic." is a strange one since I found on the Internet this title: "ANTICARIOGENIC POTENTIAL OF THE OLEORESIN EXTRACTED FROM Copaifera lucens AND ITS GENOTOXIC EFFECT"!!! Published in INVESTIGAÇÃO, 16(7):1-128, 2017 with DOI: https://doi.org/10.26843/investigacao.v16i7.2172!
You can see the attached image!
I understand the study presented in this manuscript is an older one, updated with the part related to T. gondii and, probably, more extensive regarding the anti-cariogenic effect. Anyway, the old article, already published, with slightly changed authors, is not mentioned in the references of this manuscript! I consider this aspect a major flaw, with an "air" of plagiarism, even if the published article seems to be an abstract.
Materials and Methods section
Lines 357-360: Reworded: "Between August 2012 and May 2014, we collected in Rio de Janeiro, Brazil, Copaifera lucens Dwyer oleoresin (CLO) and crude extract of Copaifera lucens (CECL) obtained from the tree leaves. The Brazilian government authorized the action through SISBIO (35143-1) and CGEN (010225/2014-5)."
Line 367: "The strains used" instead of "The strains that were used"
Line 380: "Briefly" instead of "Brefily"
Lines 384-385: Reworded: "The inoculum concentration was adjusted for each microorganism; for cariogenic bacteria, the final concentration was 5x105 CFU/mL."
Line 396: "active" or "activity"?
Line 398: cut "and" from here: "dilution, and following"
Lines 400-401: I think my rewording sounds better: "The experiments were carried out for the strains that showed better MIC results, performed in triplicates, with graphically demonstrated results".
Lines 421-422: grammatical corrections: The antibiofilm activity measured by counting the number of microorganisms was performed to assess cell viability.
Line 475/475: "...the cells; then, PA was..." instead of "...the cells; and then, PA was..."
Line 516: "was evaluated" instead of "was evaluation"
Line 550: "...PA, and the procedure..." instead of "...PA, the procedure..."
Line 552: "was used; the nematodes" instead of "was used, the nematodes". Respect the marks within a phrase!
Conclusions section
Line 591: use italic for species name: T. gondii!

The manuscript requires extensive revision.